# Research on the Improvement Effect and Mechanism of Micro-Scale Structures Treated by Laser Micro-Engraving on 7075 Al Alloy Tribological Properties

**DOI:** 10.3390/ma12040630

**Published:** 2019-02-20

**Authors:** Mingkai Tang, Lichao Zhang, Yusheng Shi, Wenzhi Zhu, Nan Zhang

**Affiliations:** School of Materials Science and Engineering, Huazhong University of Science and Technology, 1037 Luoshi Road, Wuhan 430074, China; wate8788@yahoo.com (M.T.); start919@126.com (W.Z.); pluto1347@126.com (N.Z.)

**Keywords:** Al alloy, micro-scale structure, tribological properties, effect mechanism

## Abstract

During various applications in aerospace, ships, autos, and aircraft, 7075 Al alloy will frequently contact other materials, and therefore suffer from slight abrasion. However, the poor tribological properties of 7075 Al alloy greatly affect its performance and life length, leading to limitations in its application. Preparing roughness structures on the surface is regarded as a promising method to improve the properties of materials. However, the tribological properties of 7075 Al alloy cannot be enhanced significantly by roughness structures in complex dynamic changeable environments, owing to the incomplete understanding of the effect of roughness structures. Given the above issues, in this paper, micro-scale structures (linear grooves, gridding grooves, and arc grooves) were designed and prepared on 7075 Al alloy surfaces by a surface treatment (laser micro-engraving), which provides excellent controllability of the morphology and dimensions of as-prepared roughness structures. The tribological properties of the as-prepared surfaces were investigated systemically. The effect of micro-scale structures on the tribological properties was studied. The wear mechanism and tribological properties improvement mechanism of the surfaces were clarified. Furthermore, the effect degree of the enhancement factors of the micro-scale structures on the tribological properties was explored under different conditions. The results indicate that the micro-scale structures play an important role in improving the tribological properties of Al alloy under different sliding speeds. The improvement mechanism can be summarized by four factors. However, the effect degrees of these factors on the tribological properties exhibit considerable differences. This study not only develops specific micro-scale structures that can dramatically improve the tribological properties of 7075 Al alloy under different conditions, but also offers guidance for the construction of appropriate roughness structures that can dramatically improve the tribological properties of Al alloy according to the friction conditions.

## 1. Introduction

Al alloys exhibit numerous attractive properties, such as low cost, low density, excellent thermal conductivity, being non-ferromagnetic, and great anticorrosion, leading to their extensive applications in many fields, including the automobile, sensor, aeronautical, and space fields [1,2,3,4,5]. Because of its strength, toughness, and stiffness, 7075 Al alloy has numerous applications in aerospace, ships, aircraft, and transportation. During application in the above fields, 7075 Al alloy will frequently contact other materials, and therefore suffer from slight abrasion. However, Al alloy has low hardness and poor wear resistance. Furthermore, the Al alloy has a high friction coefficient during dry sliding contact, resulting in aggravated wear and energy loss. Therefore, local destruction and reduction of performance frequently occurs with the Al alloy during its service, which weakens the performance stability and life length significantly. The applications of 7075 Al alloy are greatly affected owing to the above shortcomings [6,7]. Based on the increasing demand for Al alloys in the above fields, it is essential to improve the tribological properties of Al alloy.

Roughness structure preparation refers to constructing specific micro-scale structures, such as micro-dimples or micro-grooves, on the material’s surface, which serve as a surface modification to improve the surface properties of the material [8,9,10]. At present, there are several methods to construct micro-scale structures on the surface of Al alloys, which can be generally divided into mechanical machining, energy beam etching, and coating techniques [11,12,13,14]. In recent years, as a promising method to controllably prepare roughness structures, laser treatment has shown numerous advantages, such as high efficiency, environmental friendliness, low-cost, convenience, and facile operation [15,16,17,18]. Furthermore, it provides excellent controllability of the morphology and dimensions of as-prepared roughness structures, which can allow the achievement of the optimum design and preparation of roughness structures [19,20].

Nowadays, many studies have indicated that roughness structures can enhance tribological properties in comparison with the performance obtained with smooth surfaces [21,22,23,24]. For example, Liu et al. investigated the tribological properties of titanium roughness structure surfaces covered with bumps arranged in a random array. The results indicate that this surface exhibits a much lower friction coefficient than a smooth surface under different loads [25]. Xu et al. fabricated micro-scale linear grooves and dimples on cylinder liners. Tribological tests showed that the micro-dimple surfaces have the best antifriction and antiwear properties under lubrication with esterified bio-oil. Compared with untreated surfaces, the friction coefficient and wear rate decrease by 22% and 58%, respectively [26]. Tribological behaviors of shell-like micro-scale structure surfaces of Ti_3_AlC_2_ ceramic were studied under loads of 1, 5, and 9 N by Xu et al. [27]. They found that the tribological properties of the Ti_3_AlC_2_ ceramic surface was greatly enhanced by the micro-scale structure.

Up to now, though the tribological properties of micro-scale structure surfaces of various materials have been studied, little research has been carried out to investigate the effect of such structures on the tribological properties of Al alloys. Hu et al. [28] studied the tribological properties of the micro-dimple surface of Al–Si alloy containing MoS_2_ solid lubricant. In our previous study, the tribological properties of Al alloy micro-scale structure surfaces were only preliminarily studied [20]. However, the wear mechanism of Al alloy micro-scale structure surfaces is not clarified completely. The effect degree of micro-scale structures on the tribological properties under different conditions, and the corresponding effect mechanism, are also not clearly understood. Because of this, the tribological properties of Al alloy cannot currently be improved by micro-scale structures in the most efficient way. Under some conditions, micro-scale structures could even weaken the tribological properties of Al alloys.

In response to the aforementioned issues, in this paper, specific micro-scale structures were developed and prepared on the surface of Al alloys by a surface treatment (laser micro-engraving) that provides excellent controllability of the morphology and dimensions of as-prepared roughness structures. The tribological properties of the surfaces were investigated systemically under different dry sliding contact conditions. The effect of micro-scale structures on the tribological properties was studied. The wear mechanism and tribological properties improvement mechanism of the as-prepared surfaces were also clarified. Furthermore, the effect degree of enhancement factors of the micro-scale structures on the tribological properties, and the corresponding mechanisms, were explored.

## 2. Materials and Methods

### 2.1. Materials

The commercial AA7075 aluminum alloy (in mass percent: 0.4 Si, 1.2 Cu, 0.3 Mn, 2.1 Mg, 0.20 Cr, 5.1 Zn, and the balanced Al), purchased from Alcan Alloy Products Company Ltd. (Montreal, Canada), was chosen as the test material. The Al alloy plate was cut in the form of a 25 mm × 25 mm square pin by wire cut electrical discharge. The cut samples were polished using emery papers (from 500 to 1200 grit), to create the following treatments.

### 2.2. Surface Treatments

The samples were treated by laser micro-engraving (YAG laser, laser wavelength of 1064 nm, laser pulse duration of 20 ns, laser diameter of 80 μm, output power of 20 W) (Chuanlei Laser Equipment Ltd., Shanghai, China). Three kinds of regular micro-scale structures (linear groove, gridding groove, and arc groove) with different texture spacings were designed and fabricated on the Al alloy. The schematic of the expected morphology of the micro-scale structure surfaces is shown in Figure 1. The detailed processing parameters of the surfaces are shown in Table 1. In order to study the influence of the morphology and distribution strategy of micro-scale grooves on the tribological properties, linear grooves and gridding grooves were developed. Meanwhile, according to the testing mechanism of a ball-on-disk tribometer, the arc groove was developed to investigate the effect of the micro-scale groove orientation on the tribological properties of Al alloys. After treatment, all the samples were cleaned twice in an ultrasonic bath using alcohol, for 10 min.

### 2.3. Characterization

The morphology of the micro-scale structure surfaces was investigated by a Quanta 200 FEG environmental scanning electronic microscope (ESEM) (FEI Ltd., Hillsboro, OR, USA). Electron dispersive spectroscopy (EDS) (Oxford Instruments Ltd., Oxford, UK) was used to investigate the composition of the surface. The Vickers hardness of the as-prepared micro-scale structures (cross section) was measured using a HVS-1000 Vickers hardness instrument (Sanfeng Science and Technology Ltd., Beijing, China), with a load of 0.5 kp and a dwell time of 8 s. Eight tests were carried out and the mean value was given. The density of specimens was determined by Archimedes’ principle. The tribological properties of the samples were tested with a HT-1000 ball-on-disk tribometer (Lanzhou Institute of Chemical Physics, Lanzhou, China). The disk was the aforementioned surface modified samples treated by laser micro-engraving. The counterpart ball with a diameter of 6 mm was made of steel (GCr15, 600 HV, Ra 0.02 μm). Prior to commencing the test, the disks and counterpart balls were ultrasonically cleaned with acetone, and then thoroughly dried in hot air.

## 3. Results and Discussion

### 3.1. Characteristics of As-Prepared Micro-Scale Structure Surfaces

Figure 2 shows SEM images of the micro-scale structure surfaces with texture spacing of 150 μm, treated by laser micro-engraving. As shown in Figure 2a, regular linear grooves were constructed on the surface. The linear grooves were parallel to each other. As seen in the high-magnification SEM images shown in Figure 2b,c, the width of the convex structure and groove was about 80 μm and 70 μm, respectively. Moreover, the spacing of the linear grooves was 150 μm, which was consistent with the preset texture spacing. As shown in Figure 2d, it could be observed that a gridding groove was formed on the surface. The angle of the gridding groove was 90°. Due to the effect of the gridding groove, a square convex array was formed on the surface. The size of the square convex structure was about 80 μm × 80 μm (Figure 2e). The high-magnification SEM image shown in Figure 2f indicates that a pit was constructed at the point of the intersection of grooves. Thus, the grooves were not totally interconnected. Furthermore, the width of the grooves was 70 μm. The spacing of the grooves was the same as the preset value. In the SEM image shown in Figure 2g, regular arc grooves can be seen on the surface. Due to the same processing parameters, the morphology and geometrical dimensions of the roughness structures (arc grooves and convex structures by the side of the grooves) were similar to those of the linear grooves.

As shown in Figure 3, as the texture spacing increases to 250 μm, the morphology of the micro-scale structures of the as-prepared surfaces (linear groove surface, gridding groove surface, and arc groove surface) underwent almost no change because of having the same laser parameters. Moreover, the width of the groove of the surfaces was maintained at 70 μm. However, the width of the convex structure showed an increase with the increase of texture spacing. Due to the 100 μm increase of texture spacing, the width of the convex structure of the linear groove surface and arc groove surface rose to 180 μm. Meanwhile, the size of the square convex structure of the gridding groove surface was 180 μm × 180 μm. The results indicate that the morphology of the micro-scale structures of the as-prepared surfaces underwent almost no change with the increase of texture spacing. The width of convex structure increased corresponding to the increment of texture spacing. Therefore, as the texture spacing rose to 350 μm, the morphology of the surfaces was nearly the same. The detailed dimensions of the micro-scale structures of the surfaces with different texture spacings are shown in Table 2.

After laser micro-engraving treatment, it is considered that the chemical composition of the surface may be changed, along with the morphology. Due to its influence on the Al alloy properties, the composition is an important factor that cannot be ignored. Thus, electron dispersive spectroscopy (EDS) was carried out to investigate the chemical composition of the micro-scale structure surfaces.

The EDS energy spectrum of the micro-scale structure surfaces with a texture spacing of 150 μm is shown in Figure 4. It was found that the elements of the treated surfaces were the same as in the untreated surface. O, Cu, Zn, Mg, and Al were detected. From the element content of the surfaces exhibited in Table 3, it was found that the oxygen content of the surface underwent a significant increase after laser micro-engraving treatment, which was consistent with previous research. During laser micro-engraving treatment, pure aluminum is transformed into oxidized aluminum. However, the content of each element of the as-prepared surfaces with different morphologies were almost maintained at the same level, indicating that the material properties of the micro-scale structures were the same.

As is well known, the hardness can greatly affect the tribological properties of Al alloy. During laser micro-engraving treatment, because of the high temperature gradient, the hardness of the surfaces will be affected. Thus, the Vicker’s hardness of the as-prepared surfaces was measured. As shown in Figure 5, the hardness of the untreated surface was used for reference (185 HV). It can be seen that the hardness of the fabricated surfaces improved significantly. It is possible that the high temperature gradient of the material of the surface layer induced quenching strengthening during the laser micro-engraving treatment. There were no considerable differences in the hardness among different micro-scale structure surfaces. Though the hardness of the micro-scale structure surfaces underwent a slight increase with the increase of texture spacing, they were still within the same level.

### 3.2. Tribological Properties of the Linear Groove Surface (LGS)

In order to investigate the tribological properties of the micro-scale structure surfaces, tribological tests were carried out. Generally, 7075 Al alloy will frequently suffer from slight abrasions during use. Thus, a comparatively low applied load (0.3 N) and low sliding speed (100–400 mm/s) were selected. The curve of the dynamic friction coefficient of the linear groove surface samples (LGS) against a steel ball friction pair at a constant sliding speed of 100 mm/s and load of 0.3 N is shown in Figure 6a. The curve of the dynamic friction coefficient of the untreated surface is shown for reference. Initially, the friction coefficient of LGS with different texture spacing showed a fluctuation. This was because the irregular tiny structures on the convex structure surface of the LGS were in contact with the friction pair and became worn first. As the irregular tiny structures were worn out, the friction coefficient entered into a stable stage. Though the friction coefficient of the LGS showed a slight increase with the increment of texture spacing, the friction coefficient was lower than that of the untreated surface (0.46). Furthermore, when the test time rose to 5 min, due to the generation of black wear debris on the surface, the friction coefficient of the untreated surface increased dramatically. Figure 6b presents the wear rate of the LGS after tribological testing (test time of 10 min). It was found that the variation of the wear rate was similar to the friction coefficient. The wear rates of the LGS with different texture spacing were 1.68 ± 0.040 × 10^−3^, 1.92 ± 0.074 × 10^−3^, and 2.17 ± 0.023 × 10^−3^ mm^3^/N∙m, respectively, indicating that the wear rate of the LGS was much lower than that of the untreated surface. According to the above results, it can be concluded that the formed linear groove highly improves the tribological properties of Al alloy. To further investigate the tribological properties of the LGS, variations of the friction coefficient and wear rate of the LGS with sliding speed are shown in Figure 6c,d. It was observed that the friction coefficient and wear rate of the samples were lower than that of the untreated surface under different sliding speeds, which further indicated the improvement effect of the linear groove on the tribological properties of Al alloy. Furthermore, the friction coefficient, as well as the wear rate, was reduced by increasing sliding speed. By contrast, the difference of the friction coefficient or wear rate between each LGS with different texture spacings gradually expanded as the sliding speed increased. The results indicate that the effect of the linear groove on the tribological properties of Al alloy could be enhanced by increasing sliding speed. However, with the increase of texture spacing, the improvement effect would be weakened.

The SEM image of the worn surface of the untreated surface sample after tribological testing is shown in Figure 7a. Numerous tiny scratches, along with a few pits, appeared on the surface. The scratches were parallel to the sliding direction. Thus, it could be concluded that the wear mechanism of the Al alloy surface was mainly abrasive wear, accompanied by adhesive wear. In addition, a great number of flocculent particles which were the aforementioned black debris mainly consisting of O and Al as suggested by EDS analysis (Figure 7b) and element distribution of the worn surface (Figure 7c–i) were distributed at some locations on the worn surface. It appeared that some of wear debris that did not depart from the friction zone was crushed by the steel ball repeatedly. In the meantime, due to the frictional heat, the particles were gradually oxidized, leading to the formation of black flocculent particles.

Figure 8 shows SEM images of the worn surface of the LGS after testing. As shown in Figure 8a, the width of the worn zone was much larger than that of the strip convex structure of the LGS (texture spacing of 150 μm). Under a low sliding speed (100 mm/s), scratches and pits could be found on the worn surface, indicating that the wear mechanism was identical to that of the untreated surface. However, compared with the untreated surface, the worn surface of the LGS was smoother, and the scratches were smaller. Additionally, numerous irregular tiny wear debris could be observed in the grooves. As shown in Table 4, the composition of the worn surface and wear debris in the grooves were similar. The wear debris can be captured by the grooves, which can significantly decrease the degree of abrasive wear. Flocculent black wear can be prevented as well. Furthermore, the micro-sheet structure was seen at the edge of the strip convex structure of the worn surface. Meanwhile, sheet-like debris of a large size could be observed in the grooves. Thus, it could be concluded that the structure at the edge of the convex structure was flattened out by the steel ball. Meanwhile, because of the plough-effect of the steel ball, the flat structure was cut and fell into the groove. The results indicate that the wear at the edge of the convex structure of the LGS was not induced by the plough-effect of wear debris, but by that of the steel ball. This discovery results in a new understanding of the wear mechanism of Al alloy micro-scale structure surfaces. From the SEM image of the worn surface of the LGS presented in Figure 8b, we can see that as the texture spacing increased to 250 μm, a similar phenomenon could be seen because of the same wear mechanism. However, the wear debris in the grooves was smaller than before, indicating that the capturing debris effect of the roughness structure was weakened due to the decrease of the distribution density of the concave structure (groove) of the LGS. This led to an increase in the time that wear debris stayed at the friction zone. As shown in Figure 8c, when the sliding speed rose to 400 mm/s, the worn surface of the LGS with a texture spacing of 250 μm underwent almost no change. Thus, the wear mechanism of the LGS was mainly abrasive wear under a high sliding speed. Owing to the increase in the time that wear debris was crushed by the steel ball per unit time, the scratches on the worn surface and wear debris in the grooves became smaller. From the SEM image of the worn surface of the LGS with a texture spacing of 350 μm, shown in Figure 8d, we could see that a small amount of black flocculent wear debris could be found in the groove, which further proved that the increase of texture spacing would weaken the capturing wear debris performance of the LGS.

According to above results, the improvement mechanism of the tribological properties of the LGS of Al alloy can be summarized by the following: (1) Hardness of the surface is improved, (2) the concave structure of the surface decreases the real contact area between the surface and steel ball, and (3) the linear groove of the surface can capture the wear debris, which reduces the degree of abrasive wear. However, the effect degree of the above three factors is different under different friction conditions. Based on the above tribological test results, under a low sliding speed, the speed of the wear debris entering into the groove was low, leading to a poor capturing wear debris effect of the grooves of the LGS. Meanwhile, when compared with the effect of hardness improvement, the reduction of real contact area induced by the grooves exhibited a larger influence. Therefore, the friction coefficient and wear rate increased with the increase of texture spacing. By contrast, the efficiency of the wear debris entering into the groove underwent a significant increase with the increase of sliding speed, which lead to the improvement of the capturing wear debris effect of the grooves. Thus, as the sliding speed rose, the tribological properties of the surface were enhanced. The difference of the friction coefficient or wear rate between each LGS with different texture spacings gradually increased as well.

Figure 9 presents the SEM images of the boundary of the worn surface of the LGS. It was found that massive wear debris was accumulated in the grooves. However, the amount of wear debris in the groove away from the friction zone became less and less. Moreover, a lot of wear debris could be found in the groove outside of the friction zone. This result indicated that the grooves of the LGS cannot only captured wear debris, but also moved the wear debris away from the friction zone. It was easy for the newly-produced wear debris to enter into the grooves, which could further improve the capturing wear debris effect of the grooves of the LGS.

### 3.3. Tribological Properties of the Gridding Groove Surface (GGS)

In the following, the tribological properties of the gridding groove surface samples (GGS) were investigated. From the curve of the dynamic friction coefficient of the GGS against a steel ball friction pair (sliding speed of 100 mm/s and load of 0.3 N) shown in Figure 10a, it can be seen that the friction coefficient ranged from 0.40 to 0.42 in the stable stage. It was lower than that of the untreated surface, indicating that the gridding groove could decrease the friction coefficient of the Al alloy efficiently. Based on the above research, this was due to three factors: (1) Hardness of the surface was improved, (2) the concave structure of the GGS decreased the real contact area between the surface and the steel ball, and (3) the gridding grooves of the GGS could capture wear debris. With the increase of texture spacing from 150 to 250 μm, the friction coefficient of the GGS maintained the same level. Because of the extremely low distribution density of the convex structure of the GGS with a texture spacing of 150 μm, the deformation of the GGS was large under the applied load, leading to the increase of the real contact area between the surface and the friction pair. As the texture spacing increased, the increment of the distribution density of the convex structure of the GGS decreased the deformation degree. When the texture spacing further increased, the friction coefficient of the GGS increased, owing to the weakening of the effect of distribution density of the convex structure on the deformation degree and an increase of the real contact area between the surface and the friction pair. Figure 10b shows the wear rate of the GGS after tribological testing. Obviously, as texturing spacing increased, the variation of the wear rate of the GGS was the same as that of the LGS, which further shows the adverse influence of texture spacing. However, compared with the untreated surface sample, the GGS with different texture spacings presented with a lower wear rate, indicating that the gridding groove was able to improve the Al alloy tribological properties efficiently. As shown in Figure 10c,d, the friction coefficient and wear rate (10 min test) of the GGS with different texture spacings were much lower than that of the untreated surface sample. The results indicate that the gridding groove can enhance the tribological properties of the Al alloy under different sliding speeds. Moreover, as shown in Figure 10c, the friction coefficient of the GGS with a texture spacing of 150 μm became lower than that of the surface with a texture spacing of 250 μm. It could be speculated that the wear mechanism of the GGS was mainly abrasive wear. As the sliding speed increased, the performance of the gridding groove in capturing wear debris gradually increased, leading to the improvement of the capturing wear debris effect of the GGS on the tribological properties of the Al alloy. By contrast, according to aforementioned results, the increase of texture spacing could weaken the capturing wear debris performance of the GGS. Therefore, the difference of the friction coefficient or wear rate between each GGS increased with the increase of sliding speed.

Figure 11 shows the SEM images of the worn surface of the GGS after friction. Clearly, the area of the worn zone was much larger than the convex structure of the GGS. As shown in Figure 11a,b, under a low sliding speed (100 mm/s), numerous tiny scratches, which were parallel to the sliding direction, appeared on the worn surface, proving the above speculation. The wear mechanism of the GGS was mainly abrasive wear. Moreover, irregular particles could be found in the grooves. The chemical composition of the particles was similar to that of the worn surface, as suggested in Table 4, which further indicated that the gridding grooves could capture wear debris. According to the observation, there were some barriers in the grooves, leading to the disconnection of the grooves. Thus, it was considered that the gridding grooves could not remove wear debris away from the friction zone. In addition, the micro-sheet structure was seen at the edge of the convex structure of the GGS. Sheet debris with large sizes could be found in the grooves as well. This result further indicated that wear was mainly induced by the plough-effect of the steel ball at the edge of the convex structure. As the sliding speed increased, a similar phenomenon could be seen (Figure 11c). As shown in Figure 11d, under a high sliding speed, black flocculent wear debris appeared on the worn surface of the GGS, with a texture spacing of 350 μm. This result indicated that the capturing wear debris performance of the GGS was weakened by increasing texture spacing. Additionally, compared with the worn surface shown in Figure 8d, the amount of flocculent wear debris showed a significant increase. It is speculated that the performance of the gridding groove in dealing with wear debris is worse than that of the linear groove.

### 3.4. Tribological Properties of the Arc Groove Surface (AGS)

Figure 12a shows the friction coefficient of the arc groove surface samples (AGS) against a steel ball friction pair at a constant sliding speed of 100 mm/s and load of 0.3 N. Clearly, compared with the untreated surface, the friction coefficient of the AGS with a texture spacing of 150 μm underwent a dramatic decrease in the stable stage. The friction coefficient at this point was about 0.33. As the texture spacing increased from 150 to 350 μm, the friction coefficient of the AGS increased significantly. From the wear volume of the AGS after testing (10 min), exhibited in Figure 12b, similar variation could be found. The wear volume rose greatly with increasing texture spacing. Moreover, the friction coefficient and wear volume of the samples with different texture spacings were lower than that of the untreated surface sample, showing a great improvement effect on the tribological properties of the Al alloy.

Based on the aforementioned results, the improvement effect of the micro-scale structures on the Al alloy’s tribological properties was based on three factors. The arc convex structure and concave structure of the AGS were parallel to the sliding direction during tribological testing. As shown in Figure 13, compared with the strip concave structure that was perpendicular to the sliding direction, the concave structure of the AGS could not only decrease the real contact area, but also weaken the ploughing effect induced by the steel ball, which greatly reduced the production of wear debris. Thus, the tribological properties of the Al alloy could be dramatically improved by arc grooves. As the texture spacing increased, the aforementioned influence of the concave structure of the AGS decreased significantly, resulting in aggravation of friction and wear. In addition, owing to running in the same direction, arc grooves could not remove wear debris away from friction zone, but rather captured wear debris during the friction process.

The variations of the friction coefficient and wear rate of the AGS with sliding speed are shown in Figure 14. It was found that the friction coefficient and wear rate of the samples underwent a slight decrease with the increase of sliding speed. In the meantime, the difference of the friction coefficient or wear rate between each AGS with different texture spacing gradually increased. The results further prove that the effect of capturing wear debris on the tribological properties was enhanced greatly as the sliding speed increased. However, the capturing wear debris performance of the concave structure of micro-scale structure surfaces became worse when increasing the texture spacing.

From the SEM image of the worn surface of the AGS with a texture spacing of 150 μm, shown in Figure 15a, a great number of micro scratches could be seen on the surface. All of the scratches were parallel to the sliding direction, which proved that the orientation of the convex structure was consistent with the sliding direction throughout the friction test. Compared with the scratches revealed in Figure 8a and Figure 11a, the scratches on the worn surface were much smaller. This further indicated that the ploughing effect induced by the steel ball was weakened by the arc grooves. Additionally, the sheet wear debris in the grooves was much smaller than that in the grooves of the LGS and GGS with a texture spacing of 150 μm, indicating that the ploughing effect on the micro-sheet structure at the edge of the convex structure was diminished as well. As shown in Figure 15b, as the texture spacing increased to 250 μm, a similar phenomenon occurred. When the sliding speed rose to 400 mm/s, due to the same wear mechanism, the worn surface of the AGS with a texture spacing of 250 μm was similar to the worn surfaces that were observed previously. As can be seen in the highly-magnified SEM image of the worn surface (texture spacing of 350 μm), black flocculent wear debris appeared in the grooves, which resulted from the weakening of the effect of the micro-scale structure in dealing with wear debris.

### 3.5. Improvement Mechanism and Influence Degree of the Micro-Scale Structures

The improvement mechanism of the tribological properties of Al alloy micro-scale structure surfaces can be summarized by four factors: (1) The improvement of surface hardness prevents the material from deforming, resulting in the decrease of the real contact area between the friction pair and the surface, (2) the concave structure of micro-scale structure surface can decrease the real contact area between friction pair and the surface, (3) the concave structure of the micro-scale structure surface can capture wear debris, which reduces abrasive wear significantly, and (4) the concave structure, which has an orientation parallel to the sliding direction, can weaken the ploughing effect induced by the steel ball. To further understand the influence of as-prepared micro-scale structures on the tribological properties of Al alloy, the effect degree of the above factors on the tribological properties was investigated.

As suggested by Table 2, the distribution densities of the convex structure and the concave structure of the LGS with a texture spacing of 150 μm were almost consistent with those of the GGS with a texture spacing of 250 μm. From the friction coefficient and wear rate of two samples against a steel ball friction pair at a constant sliding speed of 100 mm/s and load of 0.3 N for 10 min, as shown in Figure 16, it was observed that, compared with the GGS, the friction coefficient and wear rate of the LGS were a little higher. From the friction coefficient and wear rate of two samples shown in Figure 16 (test conditions: steel ball friction pair, load of 0.3 N, sliding speed of 100 mm/s, 10 min), the friction coefficient of the samples was not considerably different. Nevertheless, the wear rate of LGS (1.68 ± 0.040 × 10^−3^ mm^3^/N∙m) was lower than that of GGS (1.98 ± 0.064 × 10^−3^ mm^3^/N∙m). Due to their similar surface hardness, this indicated that the capturing wear debris performance of linear grooves was much better than that of gridding grooves during the friction process. This was due to the ability of the linear groove to remove wear debris away from the friction zone. However, the difference of the friction coefficient or wear rate between the samples was not large, showing there was little effect from the capturing of wear debris on the tribological properties of the Al alloy under a low sliding speed. When the sliding speed was low, the movement of the wear debris was slow during the friction process, leading to a poor performance in capturing wear debris by the groove. Therefore, it could be concluded that the improvement effect of the micro-scale structure on the tribological properties of the Al alloy was mainly due to the reduction of the real contact area induced by a concave structure under a low sliding speed.

According to the above results presented in Figure 2, Figure 3, and Table 2, the morphology and geometrical dimensions of LGS and AGS micro-scale structures were almost the same. Figure 17 presents the friction coefficient and wear rate of the LGS and AGS against a steel ball friction pair at a constant sliding speed of 100 mm/s and load of 0.3 N for 10 min. It can be seen that the tribological properties of the AGS with a texture spacing of 150 μm were much better than those of the LGS, due to the reduction of the ploughing effect. It is speculated that the reduction of the ploughing effect induced by the arc grooves played an important role in the improvement of the tribological properties. As the texture spacing increased, the difference of the friction coefficient or wear rate between the two samples decreased significantly, which further proved this was likely to be the case.

The friction coefficient and wear rate of the micro-scale structure surface samples with different texture spacings under different sliding speeds are exhibited in Figure 18 (friction pair of steel balls, load of 0.3 N, friction time of 10 min, room temperature). Obviously, the friction coefficient of the AGS was the lowest, showing the influence of the weakening of the ploughing effect on the tribological properties (Figure 18a–c). Furthermore, as shown in Figure 18a, though the distribution density of the concave structure of the GGS (150 μm) was much lower than that of the LGS (150 μm), the LGS had a lower friction coefficient. This result further indicated that the effect of micro-scale structure with low distribution densities (less than 40%) on the tribological properties was mainly due to the improvement of surface hardness. However, when the distribution density of the concave structure of the micro-scale structure surface was high (more than 40%), due to the improvement of the effect of decreasing the real contact area induced by the concave structure, the friction coefficient of the GGS was lower than that of the LGS (Figure 18b,c). As the sliding speed increased, the friction coefficient of the LGS showed a significant decrease, and rapidly approached the value of the AGS. This indicated that an increase of sliding speed could improve the performance and effect of capturing wear debris of the concave structure of the micro-scale structure surface, which could remove wear debris away from the friction zone very effectively. From the variations of the wear rate of the micro-scale structure surface samples with the sliding speed shown in Figure 18d–f, a similar phenomenon could be seen. Clearly, the difference of the wear rate between the AGS and LGS reduced dramatically with the increase of sliding speed, which further proved that the capturing wear debris performance of the concave structure was affected by the sliding speed. Moreover, it indicated that the LGS had better tribological properties than other surfaces under a high sliding speed.

## 4. Conclusions

Specific micro-scale structures that can dramatically improve the tribological properties of 7075 Al alloy under different sliding speeds were developed. The tribological properties of the micro-scale structure surfaces of the Al alloy (linear groove surface, gridding groove surface, and arc groove surface), prepared by laser micro-engraving treatment, were investigated systemically. The wear mechanism and tribological properties improvement mechanism of the as-prepared surfaces were also clarified. Furthermore, the effect degree of the enhancement factors of the micro-scale structures on the tribological properties, and the corresponding mechanism, were explored. The main findings were as follows:The as-prepared micro-scale structure surfaces can greatly improve the tribological properties of 7075 Al alloy. The tribological properties improvement mechanism of Al alloy micro-scale structure surfaces can be summarized by four factors: (1) The improvement of surface hardness decreases deformation of the surface layer materials at a constant load, and improves resistance to abrasive wear, (2) the concave structure of the surface reduces the real contact area between the friction pair and the surface, (3) the concave structure can capture wear debris and reduce abrasive wear, and (4) the concave structure, which has an orientation parallel to the sliding direction, can weaken the ploughing effect induced by the steel ball.The concave structure of the gridding groove surface and arc groove surface can only capture wear debris. However, the concave structure of the linear groove surface can not only capture wear debris, but also move wear debris away from the friction zone, which further improves the performance of the capturing wear debris effect and decreases the abrasive wear of Al alloy.Due to the significant influence of the concave structure distribution density of the micro-scale structure surface on reduction of the real contact area and performance of capturing wear debris, the effect of the micro-scale structure surface on improvement of the tribological properties of the Al alloy is weakened greatly by increasing texture spacing, under different conditions.Under a low sliding speed, if the sliding speed is parallel to the orientation of the concave structure, the reduction of the ploughing effect plays a major role in the improvement of the tribological properties. Otherwise, the improvement of surface hardness (if the concave structure distribution density is less than 40%) and the decrease of the real contact area induced by the concave structure (if the concave structure distribution density is more than 40%) play a major role in the improvement of the tribological properties.Under a high sliding speed, as the effect of capturing wear debris on the tribological properties gradually increases, the effect of the micro-scale structure on the improvement of the tribological properties is mainly due to the ability induced by the concave structure of the micro-scale structure surface of capturing wear debris.Compared with a concave structure of the micro-scale structure surface that can only capture wear debris, the capturing wear debris performance of a concave structure that can capture and remove wear debris is affected greatly by the sliding speed.Under a low sliding speed, the arc groove pattern with a texture spacing of 150 μm is the optimal choice to efficiently improve the tribological properties of 7075 Al alloy. By contrast, under a high sliding speed, the linear groove pattern with a texture spacing of 150 μm enhances the tribological properties of Al alloy dramatically.

## Figures and Tables

**Figure 1 materials-12-00630-f001:**
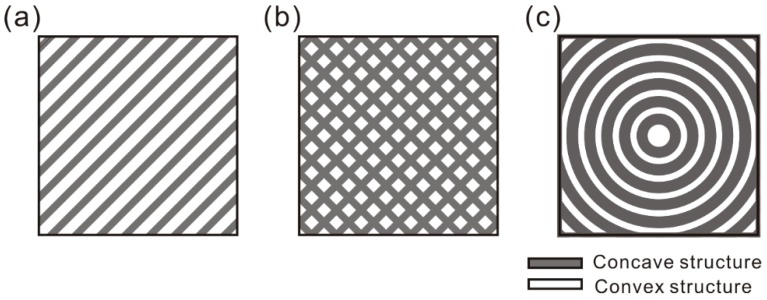
Schematic of the expected morphology of the micro-scale structure surfaces: (**a**) linear groove, (**b**) gridding groove, (**c**) arc groove.

**Figure 2 materials-12-00630-f002:**
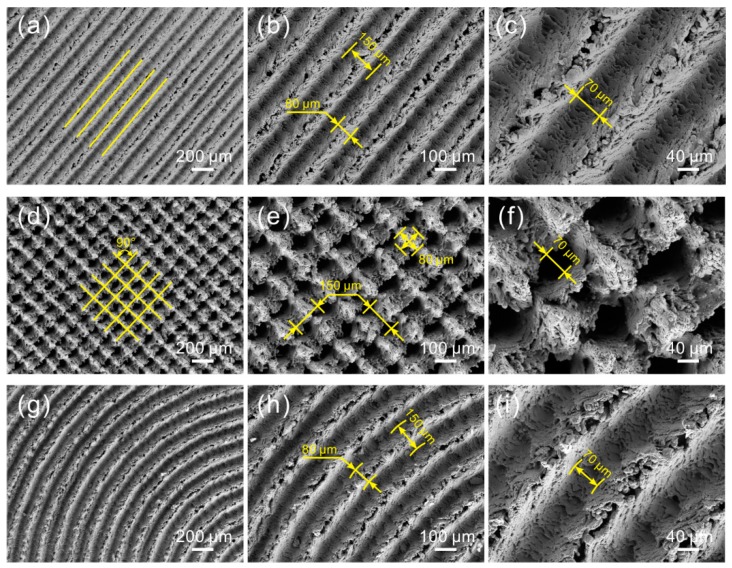
SEM images of the morphology of the Al alloy micro-scale structure surfaces (texture spacing of 150 μm): (**a**) linear groove, (**d**) gridding groove, (**g**) arc groove; (**b**,**c**,**e**,**f**,**h**,**i**) corresponding high-magnification SEM images.

**Figure 3 materials-12-00630-f003:**
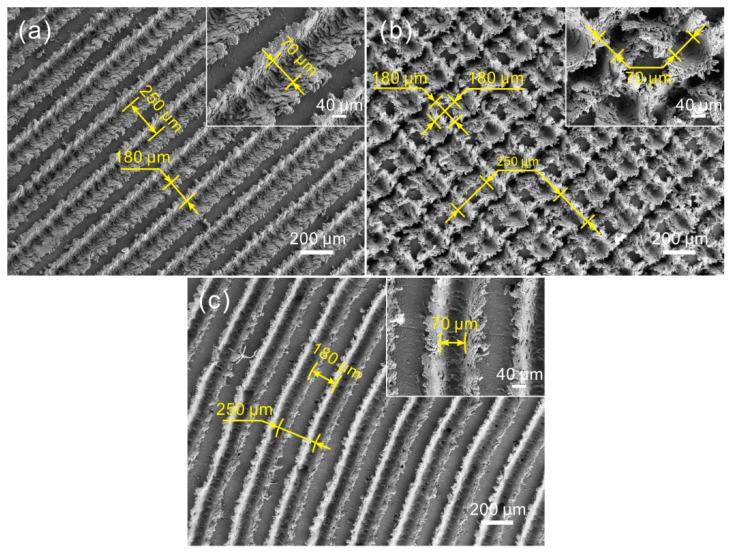
SEM images of the morphology of the micro-scale structure surfaces (texture spacing of 250 μm): (**a**) linear grooves, (**b**) gridding groove, (**c**) arc grooves (the insets correspond to their high-magnification SEM images).

**Figure 4 materials-12-00630-f004:**
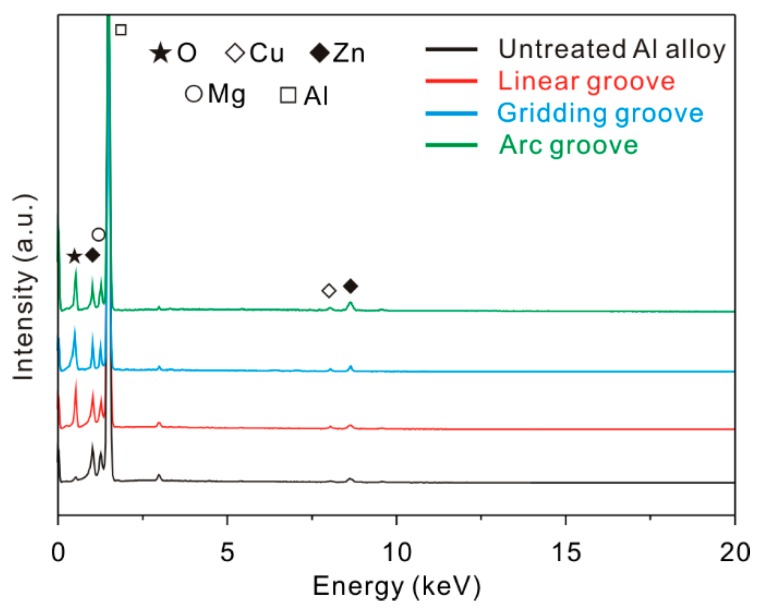
Electron dispersive spectroscopy (EDS) spectra of the Al alloy micro-scale structure surfaces.

**Figure 5 materials-12-00630-f005:**
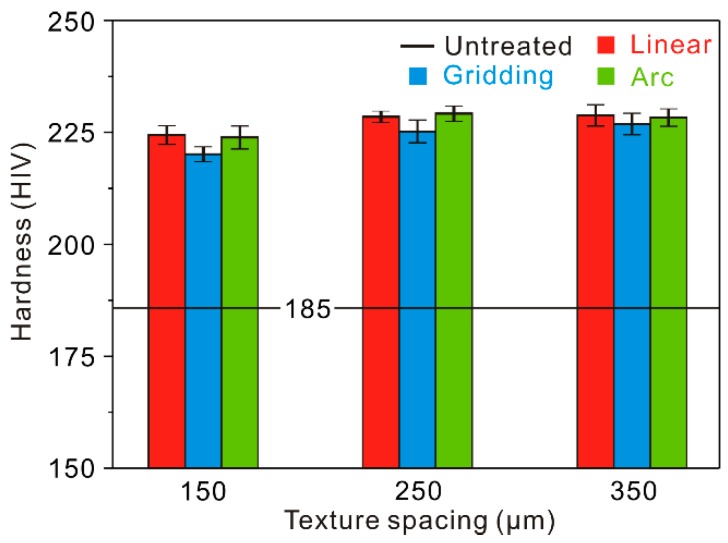
Micro-hardness of the micro-scale structure surfaces.

**Figure 6 materials-12-00630-f006:**
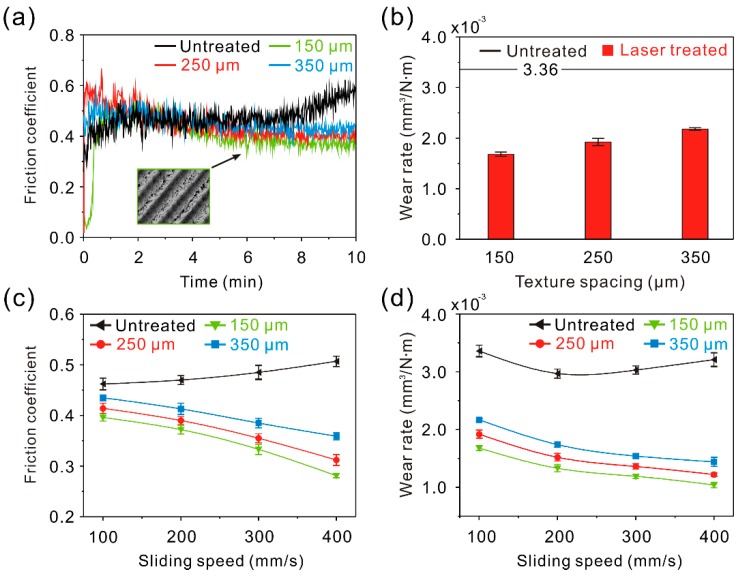
(**a**) Curve of dynamic friction coefficient of the LGS; (**b**) wear rate of the LGS; (**c**) variation of the friction coefficient of the LGS with sliding speed; (**d**) variation of the wear rate of the LGS with sliding speed (load of 0.3 N, sliding speed of 100 mm/s, frictional pair of GCr15 bearing steel, friction time of 10 min).

**Figure 7 materials-12-00630-f007:**
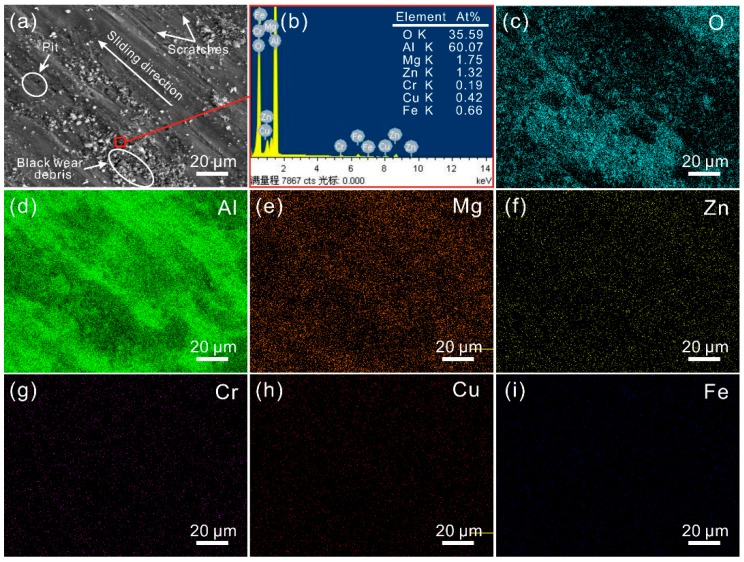
(**a**) SEM image of the worn surface of untreated surface sample; (**b**) EDS spectra of black wear debris; (**c**–**i**) element distribution of the worn surface (load of 0.3 N, sliding speed of 100 mm/s, frictional pair of GCr15 bearing steel).

**Figure 8 materials-12-00630-f008:**
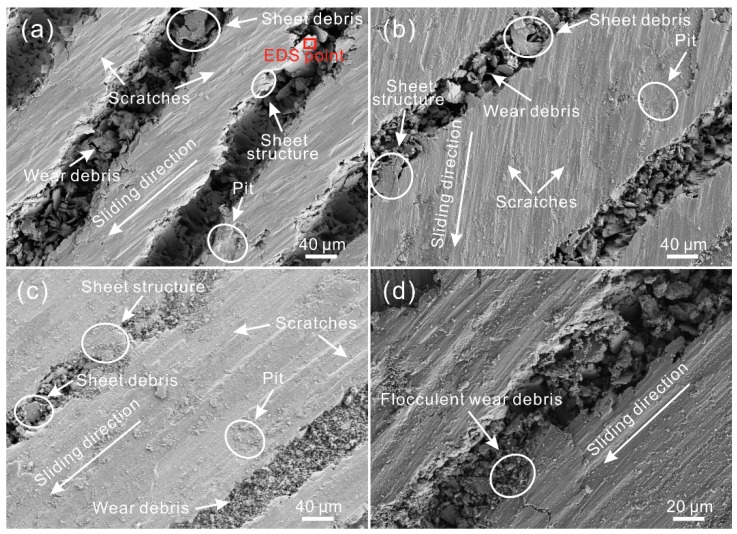
SEM images of the worn surface of the LGS with texture spacing of (**a**) 150 μm, (**b**) 250 μm (load of 0.3 N, sliding speed of 100 mm/s, frictional pair of GCr15 bearing steel); (**c**) SEM image of the worn surface of the LGS with texture spacing of 250 μm (sliding speed of 400 mm/s); (**d**) SEM image of black wear debris on the worn surface of the LGS with texture spacing of 350 μm (sliding speed of 400 mm/s).

**Figure 9 materials-12-00630-f009:**
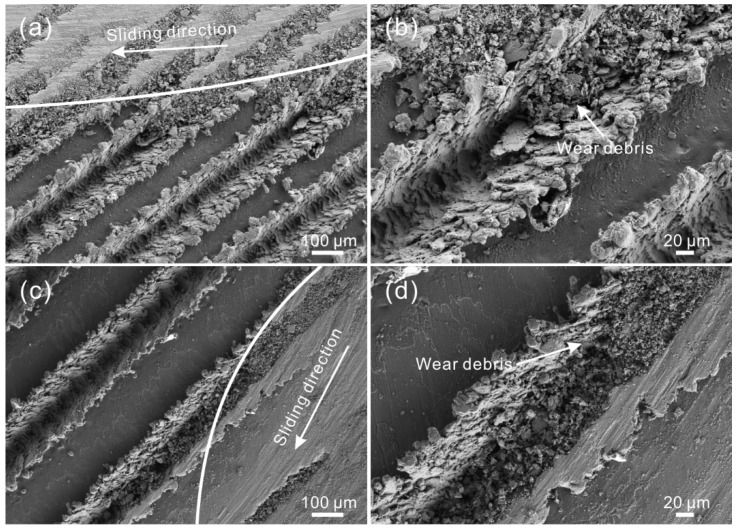
SEM image of the boundary of the worn surface of the LGS with texture spacing of (**a**,**b**) 150 μm; (**c**,**d**) 250 μm (load of 0.3 N, sliding speed of 400 mm/s, frictional pair of GCr15 bearing steel).

**Figure 10 materials-12-00630-f010:**
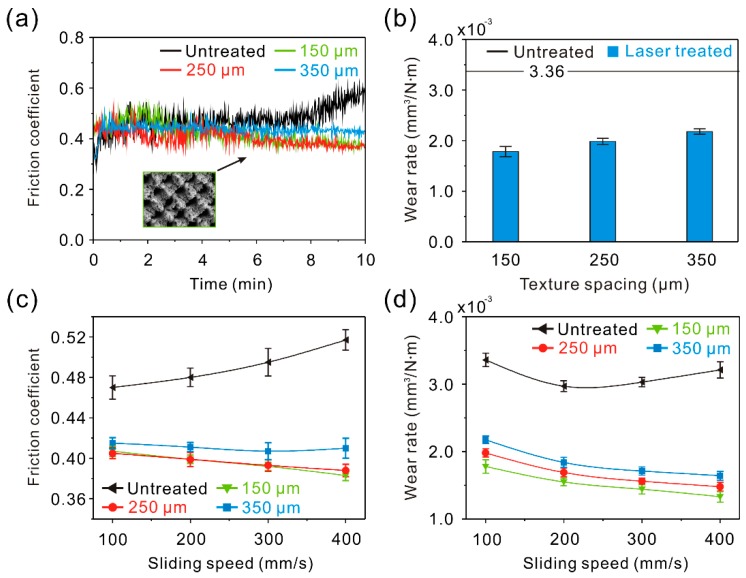
(**a**) Curve of the dynamic friction coefficient of the GGS; (**b**) wear rate of the GGS; (**c**) variation of the friction coefficient of the GGS with sliding speed; (**d**) variation of the wear rate of the GGS with sliding speed (load of 0.3 N, sliding speed of 100 mm/s, frictional pair of GCr15 bearing steel, friction time of 10 min).

**Figure 11 materials-12-00630-f011:**
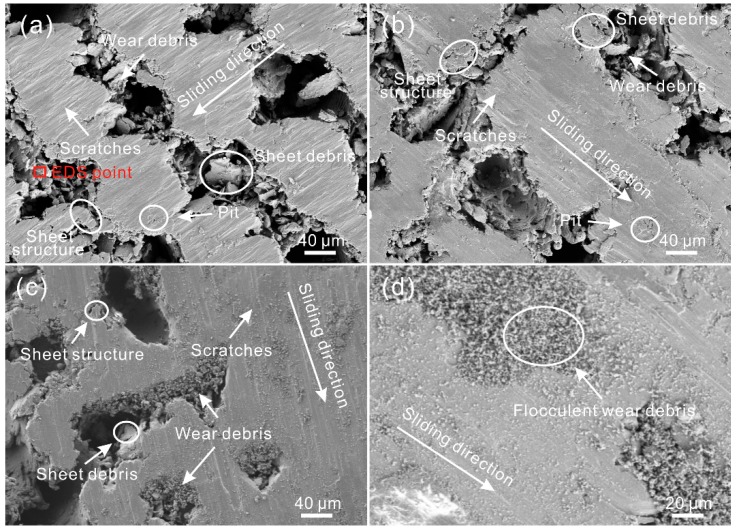
SEM images of the worn surface of the GGS with a texture spacing of (**a**) 150 μm, (**b**) 250 μm (load of 0.3 N, sliding speed of 100 mm/s, frictional pair of GCr15 bearing steel); (**c**) SEM image of the worn surface of the GGS with a texture spacing of 250 μm (sliding speed of 400 mm/s); (**d**) SEM image of black wear debris on the worn surface of the GGS with a texture spacing of 350 μm (sliding speed of 400 mm/s).

**Figure 12 materials-12-00630-f012:**
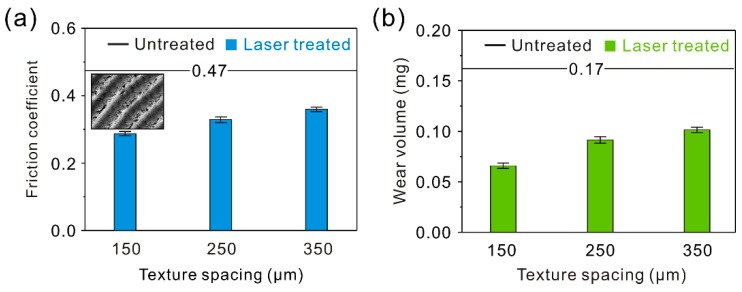
(**a**) Curve of the dynamic friction coefficient of the AGS with different texture spacings; (**b**) wear volume of the AGS with different texture spacings (load of 0.3 N, sliding speed of 100 mm/s, frictional pair of GCr15 bearing steel, friction time of 10 min).

**Figure 13 materials-12-00630-f013:**
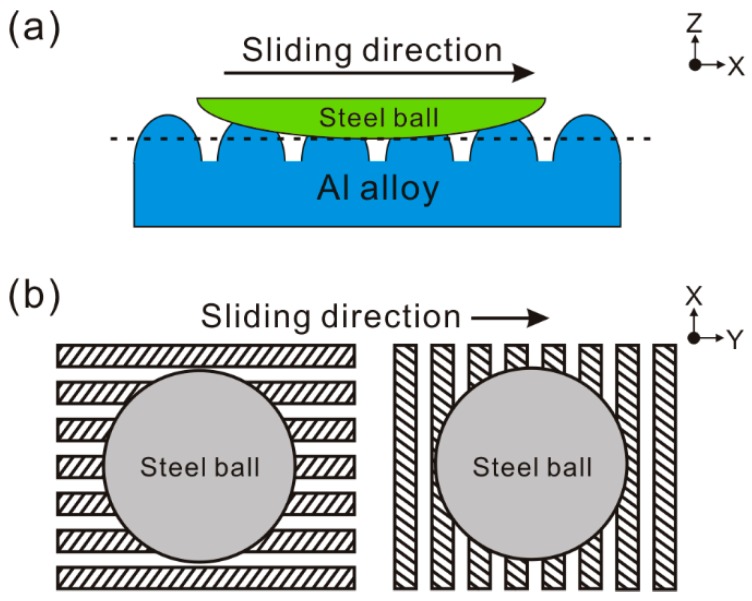
Schematic of the ploughing effect model of micro-scale structure surfaces.

**Figure 14 materials-12-00630-f014:**
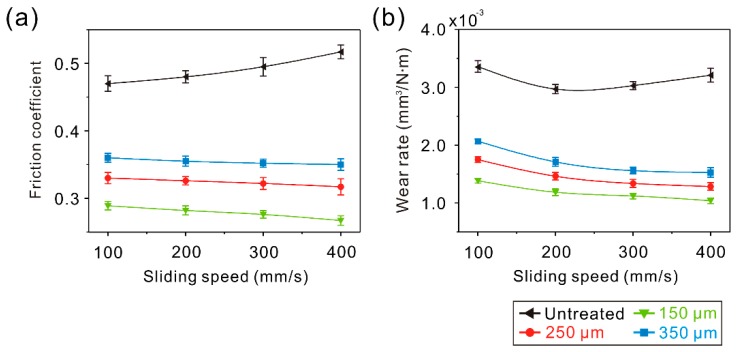
Variation of (**a**) the friction coefficient and (**b**) the wear rate of the AGS with sliding speed (load of 0.3 N, sliding speed of 100 mm/s, frictional pair of GCr15 bearing steel, friction time of 10 min).

**Figure 15 materials-12-00630-f015:**
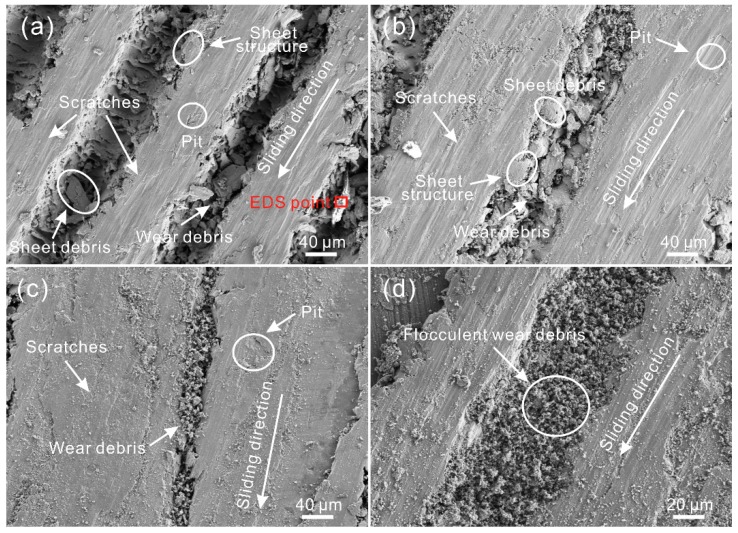
SEM images of the worn surface of the AGS with a texture spacing of (**a**) 150 μm, (**b**) 250 μm (load of 0.3 N, sliding speed of 100 mm/s, frictional pair of GCr15 bearing steel); (**c**) SEM image of the worn surface of the arc groove surface with a texture spacing of 250 μm (sliding speed of 400 mm/s); (**d**) SEM image of black wear debris on the worn surface of the AGS with a texture spacing of 350 μm (sliding speed of 400 mm/s).

**Figure 16 materials-12-00630-f016:**
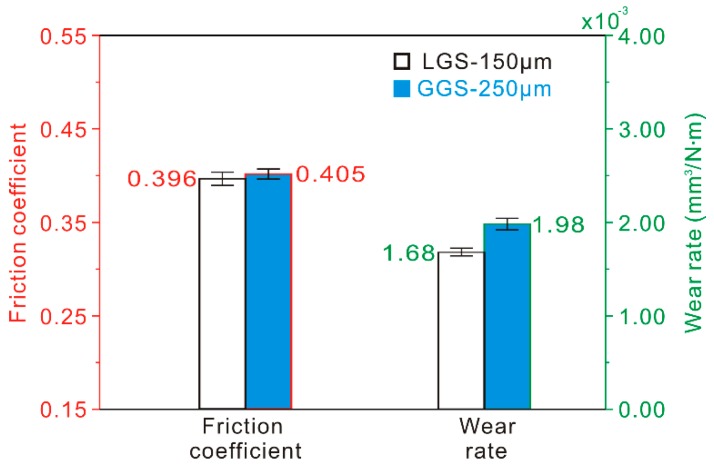
Friction coefficient and wear rate of the LGS with a texture spacing of 150 μm and GGS with a texture spacing of 250 μm (load of 0.3 N, sliding speed of 100 mm/s, frictional pair of GCr15 bearing steel, friction time of 10 min).

**Figure 17 materials-12-00630-f017:**
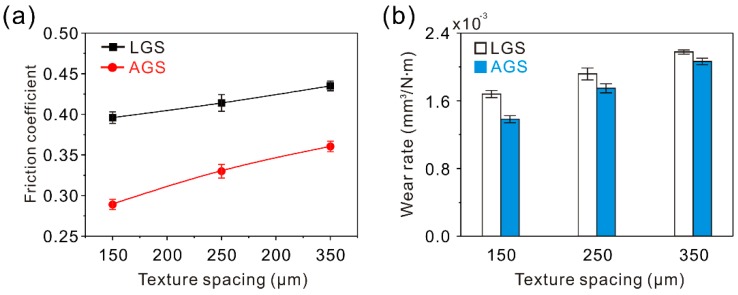
(**a**) Friction coefficient and (**b**) wear rate of the LGS and AGS with different texture spacings (load of 0.3 N, sliding speed of 100 mm/s, frictional pair of GCr15 bearing steel, friction time of 10 min).

**Figure 18 materials-12-00630-f018:**
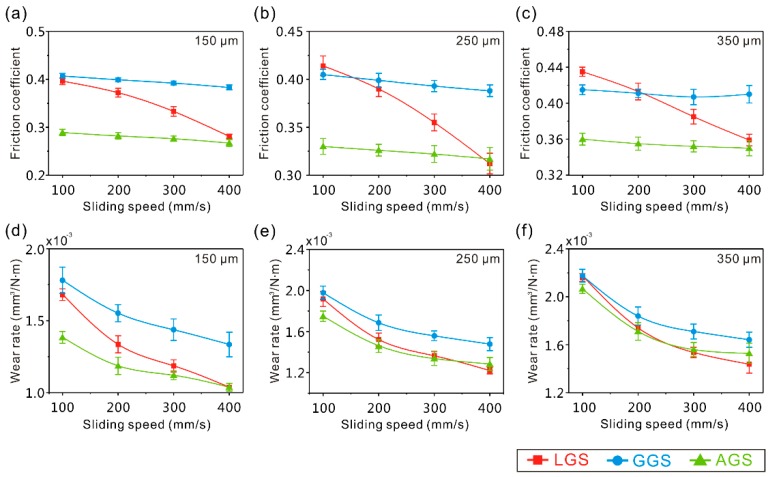
(**a**–**c**) Variations of the friction coefficient of the micro-scale structure surface samples with sliding speed; (**d**–**f**) variations of the wear rate of the micro-scale structure surface samples with sliding speed (load of 0.3 N, frictional pair of GCr15 bearing steel, friction time of 10 min).

**Table 1 materials-12-00630-t001:** Machining parameters.

Morphology	Texture Spacing (μm)			Laser Frequency (Hz)	Scanning Speed (cm/s)
Linear groove	150	250	350	5000	20
Gridding groove	150	250	350	5000	20
Arc groove	150	250	350	5000	20

**Table 2 materials-12-00630-t002:** Dimensions of micro-scale structures of the as-prepared surfaces.

**Linear Groove Surface (LGS)**	**Spacing of Linear Groove (μm)**	**Width of Convex Structure (μm)**	**Width of Concave Structure (μm)**	**Distribution Density of Convex Structure**
	150	80	70	53%
	250	180	70	72%
	350	280	70	80%
**Gridding Groove Surface (GGS)**	**Spacing of Gridding Groove (μm)**	**Size of Square Convex Structure (μm)**	**Width of Concave Structure (μm)**	**Distribution Density of Square Convex Structure**
	150	80 × 80	70	28.4%
	250	180 × 180	70	52%
	350	280 × 280	70	64%
**Arc Groove Surface (AGS)**	**Spacing of Arc Groove (μm)**	**Width of Convex Structure (μm)**	**Width of Concave Structure (μm)**	**Distribution Density of Convex Structure**
	150	70	70	53%
	250	180	70	72%
	350	280	70	80%

**Table 3 materials-12-00630-t003:** Composition content of the micro-scale structure surfaces.

Element	Untreated Surface (at %)	Linear Groove Surface (at %)	Gridding Groove Surface (at %)	Arc Groove Surface (at %)
O	1.37	11.54	11.79	11.31
Mg	2.20	3.16	3.32	3.28
Al	93.67	81.06	80.44	81.22
Cu	0.39	0.88	0.97	0.91
Zn	2.37	3.36	3.48	3.28

**Table 4 materials-12-00630-t004:** Composition content of the worn surface and wear debris in the groove measured by point EDS (texturing spacing of 150 μm, load of 0.3 N, sliding speed of 100 mm/s, frictional pair of GCr15 bearing steel).

Sample	Location	O	Mg	Al	Cu	Zn
LGS	Worn surface (at %)	7.46	2.52	86.73	0.64	2.65
	Wear debris (at %)	10.50	2.62	83.84	0.61	2.43
GGS	Worn surface (at %)	7.33	2.59	87.41	0.49	2.18
	Wear debris (at %)	9.62	2.38	85.01	0.78	2.21
AGS	Worn surface (at %)	7.95	2.76	85.97	0.74	2.58
	Wear debris (at %)	11.02	2.59	83.45	0.61	2.33

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
