# Peer review of "Research on the Improvement Effect and Mechanism of Micro-Scale Structures Treated by Laser Micro-Engraving on 7075 Al Alloy Tribological Properties"

_materials, 2019, doi:10.3390/ma12040630_

Round 1
Reviewer 1 Report
Although I have a reservation on the significance on the topic because such treatments are considered more beneficial for lubricated wear situations, the manuscript presents an interesting phenomenon of obtaining the best tribological properties with LGS. However, there's a points for improvement to the manuscript if the work to be published. Please find the comments below:
- English terms, grammar and syntax in the manuscript must be improved. A help from a native English speaker is highly recommended.
- Usually the average of 7-10 readings are enough to consider the hardness results are "reliable". Using the average of only 4 readings is not enough. The standard deviation values also need to be included.
Despite my big concerns about the reliability of the ball on desk process in quantifying wear rates, and if there's no room for the authors to select another test, I have some major comments on the tribological testing part in the manuscript:
- Running the ball on desk test for only 10 minutes does not give a clear representation for the performance of the treated samples. If one of the objective of the work is to improve the service life of 7075 Al alloys, then longer testing durations are essential.
- The authors claim that the main wear mode in the current work is abrasive wear which is not accurate. The interaction mode between the ball and specimens starts as a metal to metal (two-body). During this period, debris as a result of micro/macro fracture is going to start forming which might lead to abrasive wear (three-body). But, the authors claim that all debris get caught in the grooves. Hence, the wear mode is most likely Not abrasive wear. On the other hand, there might be an abrasive wear component in the untreated samples.
- Error bars need to be included in all graphs.
- (a) and (b) labels need to be included in Figure 7. Better image of the EDS results clearly showing all elements is needed too. On the figure caption, EPMA should be replaced by EDS. This is a huge scientific mistake and needs to be fixed.
Examples to improve English in the manuscript include:
- Line 44, the sentence "It extremely limits the further applications of ..." does not sound correct and needs to be modified.
-Line 51 and 52, "In recently decade, laser treatment is a fairly new ...is grammatically incorrect and needs to be modified.
- Line 140: "interconnected totally" should be "totally interconnected".
- Line 228" "mainly abrasive wear, accompanied by abrasive wear" does not make sense. Please revise.
- Line 249: "wear abrasive" meant to be "abrasive wear"?
In conclusion, the current work cannot be accepted at the current form. Major revisions to the work and the manuscript are needed before considering it for publishing.
Author Response
Point 1: English terms, grammar and syntax in the manuscript must be improved. A help from a native English speaker is highly recommended.
Response 1: Thank you for your comment. English terms, grammar and syntax in the manuscript have been re-edited by our native English speaking colleague and MDPI language editing service.
Point 2: Usually the average of 7-10 readings are enough to consider the hardness results are "reliable". Using the average of only 4 readings is not enough. The standard deviation values also need to be included.
Response 2: Thank you for your suggestion. The hardness was measured repeatedly 8 times at different places on the samples. The standard deviation values also exhibited in the graph.
Point 3: Running the ball on desk test for only 10 minutes does not give a clear representation for the performance of the treated samples. If one of the objective of the work is to improve the service life of 7075 Al alloys, then longer testing durations are essential.
Response 3: As you said, running the ball on desk test for only 10 minutes does not give a clear representation for the performance of the treated samples. Longer testing durations are needed. However, on one hand, there's not enough room for us to select another test. On other hand, the major objective of this work is to clarify the effect mechanism of micro-scale structures on tribological properties of 7075 Al alloy and the effect degree of enhancement factors of the micro-scale structure on the tribological properties under different dry sliding contact conditions.
Point 4: The authors claim that the main wear mode in the current work is abrasive wear which is not accurate. The interaction mode between the ball and specimens starts as a metal to metal (two-body). During this period, which might lead to abrasive wear (three-body). But, the authors claim that all debris get caught in the grooves. Hence, the wear mode is most likely not abrasive wear. On the other hand, there might be an abrasive wear component in the untreated samples.
Response 4: In the manuscript, we mention that the capturing debris performance of the micro-scale structure surfaces with various texture spacing is distinct. Thus, it is believed that it is impossible for all debris to be caught in the grooves. During the friction test, debris as a result of micro/macro fracture is formed continuously. Most of debris will be caught in the grooves. But residual and newly-produced wear debris will stays in the interface between the ball and sample, leading to abrasive wear (three-body). Furthermore, according to the worn surface of the sample shown in Figure 8, 11 and 15, numerous tiny scratches can be observed, which further indicates that the main wear mode in the current work is abrasive wear. As shown in Figure 7, a great number of scratches along with sliding direction can be found on the worn surface of the untreated sample, indicating that the wear mechanism of untreated surface is mainly abrasive wear.
Point 5: Error bars need to be included in all graphs.
Response 5: Thank you for your suggestion. Error bars were added in all graphs.
Point 6: (a) and (b) labels need to be included in Figure 7. Better image of the EDS results clearly showing all elements is needed too. On the figure caption, EPMA should be replaced by EDS. This is a huge scientific mistake and needs to be fixed.
Response 6: (a) and (b) labels have been added in the graph in the revised manuscript. EPMA has been replaced by EDS. Moreover, better image of the EDS results is showing in Figure 7.
Point 7: Line 44, the sentence "It extremely limits the further applications of ..." does not sound correct and needs to be modified.
Response 7: Thank you for your suggestion. The sentence has been modified in the revised manuscript.
Point 8: Line 51 and 52, "In recently decade, laser treatment is a fairly new ...is grammatically incorrect and needs to be modified.
Response 8: The sentence has been modified in the revised manuscript.
Point 9: Line 140: "interconnected totally" should be "totally interconnected".
Response 9: Thank you for your suggestion. The mistake has been revised in the revised manuscript.
Point 10: Line 228" "mainly abrasive wear, accompanied by abrasive wear" does not make sense. Please revise.
Response 10: Thank you for your suggestion. This is a mistake. The sentence has been revised.
Point 11: Line 249: "wear abrasive" meant to be "abrasive wear"?
Response 11: Yes, actually, "wear abrasive" meant to be "abrasive wear". This mistake has been revised in the revised manuscript.
Reviewer 2 Report
The authors reported the wear properties of surface modified Al7075 alloys by laser engraving. The wear properties were investigated against the steel ball on three different structures such as linear, gridding and Arc grove surfaces. The improvement in the tribological properties are investigated by considering the capturing wear debris of concaved regions in the GGS and AGS structures and the ability of the LGS removing the wear debris, and the simultaneous improvement in the hardness. The parameter of sliding speed is also considered. The authors made a great effort to explain the findings and discussing them. However, the article is written in poor English. The usage of long sentence constructions with more punctuations and improper usage of prepositions in some places made the article difficult to read. The article can be published provided the quality of English can be improved significantly.
Some comments given below.
1. The authors performed EDS measurements on the debris to compare the composition of the debris with the starting material. As reported by authors, there is not much change in the compositions. Since the interaction volume of EDS measurement is usually much larger, the authors need to justify that the EDS signal is not coming from the matrix beneath the debris or from the neighbouring material. Usually line scans across the point of interest is preferred to reduce the interaction volume. Point EDS measurements usually have much higher interaction volumes. More information on EDS measurements and justification needed.
2. The title of the article reads ‘Research on Improvement Effect Mechanism…’ It is unclear what the ‘Improvement Effect Mechanism’ actually means. Please clarify… If possible please provide citations addressing similar terminology.
Some of the grammatical errors are listed below and similar instances in the manuscript needs to be corrected.
Line 57: ‘researches’ should be replaced with a more suitable word such as ‘studies’.
Line 76. ‘Thus, nowadays,…’ Please rewrite the sentence. Not clear what authors mean by ‘nowadays’ in this instance.
Line 79: ‘These result that….’ What does it mean? Please rephrase the sentence.
Line 102: ‘The samples was…’. It should be ‘The samples were..’
Line 120: ‘The disk was the as-prepared samples.’ – Please rephrase and specify what the as-prepared (surface modified sample?) sample is.
Line 157: ‘… have an increase….’. Improper usage of ‘have an increase’. Please rephrase.
Line 173: ‘During laser marking treated’ or ‘treatment?’
Lines 185 – 188: Error bars should be provided for the hardness measurements. Since the treated surfaces were rough, and if the error bars are provided, one would see that there is no considerable differences in the hardness among different grooving types. Therefore, authors should justify their statement ‘the hardness of the arc groove surface was highest of all’, that one type of grove hardened the surface more than the other. How significant is the difference.
Figure 6b: The ‘sample’ in legend should be changed to ‘Laser treated?’ Please consider modifying all the instances in different figures.
Line 199: ‘…exhibited for reference’ should be replaced with ‘… shown for reference’
Line 276 - 277: rephrase the sentence ‘Therefore, as the texture spacing increases, though the hardness of the surface has a raise, the friction coefficient and wear rate increase’. Especially the word – ‘has a raise’
Capturing wear debris and trapping wear debris (Line 279) has been used in different places. It is recommended to use the standardized notation for consistency.
Line 280: ‘the difference of friction coefficient and wear rate’ – Not clear if the authors mean to subtract the friction coefficient and wear rate or is it the improper use of ‘preposition - of’. I would recommend to use ‘between’ instead.
Line 286-288: improper use of ‘…where was…’: rephrase the sentence.
Line 411-413: ‘The concave structure of micro-scale structure surface…’. It is hard to follow the sentence construction. Please rephrase.
Line 424 – 426: The sentence is very long and confusing. I think the authors meant to say that “the average friction coefficient and wear rate of LGS is higher than GGS”. Please simplify.
Line 440-442: Do the authors mean “the morphology and geometrical dimensions of LGS and AGS micro-scale structures are almost the same”?
Author Response
Point 1: The authors performed EDS measurements on the debris to compare the composition of the debris with the starting material. As reported by authors, there is not much change in the compositions. Since the interaction volume of EDS measurement is usually much larger, the authors need to justify that the EDS signal is not coming from the matrix beneath the debris or from the neighbouring material. Usually line scans across the point of interest is preferred to reduce the interaction volume. Point EDS measurements usually have much higher interaction volumes. More information on EDS measurements and justification needed.
Response 1: Thank you for your suggestion. Point EDS measurements were performed on the debris that was in the groove. Though point EDS measurements usually have much higher interaction volumes, the measurement depth of EDS is less than 10 μm. According to the SEM image of the worn surface of LGS, GGS and AGS, it can be found that the depth of the groove is at least more than 30 μm. Based on the accumulation degree of wear debris in the grooves shown in Figure 8a, 11a and 15a, it is believed that the accumulation height of wear debris in the grooves that is accumulated massive wear debris is at least more than 20 μm. Thus, the EDS signal cannot coming from the matrix beneath the debris. Moreover, the region of point EDS measurement is about 5 μm×5 μm. The effect of neighbouring material on EDS measurement result is avoid. The selected EDS measurements regions were marked in the graphs (Figure 8a, 11a and 15a) to justify that the EDS results of wear debris are not affected by the matrix beneath the debris and the neighbouring material.
Point 2: The title of the article reads ‘Research on Improvement Effect Mechanism…’ It is unclear what the ‘Improvement Effect Mechanism’ actually means. Please clarify… If possible please provide citations addressing similar terminology.
Response 2: Actually, the improvement effect mechanism means the effect of the as-prepared micro-scale structure on improving tribological properties of 7075 Al alloy and corresponding mechanism. The title has been modified in the revised manuscript.
Point 3: Line 57: ‘researches’ should be replaced with a more suitable word such as ‘studies’.
Response 3: Thank you for your suggestion. ‘researches’ has been replaced with a more suitable word in the revised manuscript.
Point 4: Line 76. ‘Thus, nowadays,…’ Please rewrite the sentence. Not clear what authors mean by ‘nowadays’ in this instance.
Response 4: The sentence has been rewritten in the revised manuscript.
Point 5: Line 79: ‘These result that….’ What does it mean? Please rephrase the sentence.
Response 5: The sentence has been modified in the revised manuscript.
Point 6: Line 102: ‘The samples was…’. It should be ‘The samples were..’
Response 6: The mistake has been corrected in the revised manuscript.
Point 7: Line 120: ‘The disk was the as-prepared samples.’ – Please rephrase and specify what the as-prepared (surface modified sample?) sample is.
Response 7: The sentence has been rephrased. The as-prepared samples were specified.
Point 8: Line 157: ‘… have an increase….’. Improper usage of ‘have an increase’. Please rephrase.
Response 8: The sentence has been rephrased in the revised manuscript.
Point 9: Line 173: ‘During laser marking treated’ or ‘treatment?’
Response 9: The mistake has been corrected in the revised manuscript.
Point 10: Lines 185 – 188: Error bars should be provided for the hardness measurements. Since the treated surfaces were rough, and if the error bars are provided, one would see that there is no considerable differences in the hardness among different grooving types. Therefore, authors should justify their statement ‘the hardness of the arc groove surface was highest of all’, that one type of grove hardened the surface more than the other. How significant is the difference.
Response 10: Thank you for your suggestion. Error bars were provided for the hardness measurements. The results of the surface hardness were re-discussed in the revised manuscript.
Point 11: Figure 6b: The ‘sample’ in legend should be changed to ‘Laser treated?’ Please consider modifying all the instances in different figures.
Response 11: Thank you for your suggestion. The graphs were modified.
Point 12: Line 199: ‘…exhibited for reference’ should be replaced with ‘… shown for reference’
Response 12: ‘…exhibited for reference’ has been replaced with ‘… shown for reference’ in the revised manuscript.
Point 13: Line 276 - 277: rephrase the sentence ‘Therefore, as the texture spacing increases, though the hardness of the surface has a raise, the friction coefficient and wear rate increase’. Especially the word – ‘has a raise’
Response 13: The sentence has been rephrased in the revised manuscript.
Point 14: Capturing wear debris and trapping wear debris (Line 279) has been used in different places. It is recommended to use the standardized notation for consistency.
Response 14: The description has been modified and use the standardized notation for consistency.
Point 15: Line 280: ‘the difference of friction coefficient and wear rate’ – Not clear if the authors mean to subtract the friction coefficient and wear rate or is it the improper use of ‘preposition - of’. I would recommend to use ‘between’ instead.
Response 15: Thank you for your suggestion. The sentence has been modified to disambiguate.
Point 16: Line 286-288: improper use of ‘…where was…’: rephrase the sentence.
Response 16: The sentence has been rephrased in the revised manuscript.
Point 17: Line 411-413: ‘The concave structure of micro-scale structure surface…’. It is hard to follow the sentence construction. Please rephrase.
Response 17: The sentence has been rephrased in the revised manuscript.
Point 18: Line 424 – 426: The sentence is very long and confusing. I think the authors meant to say that “the average friction coefficient and wear rate of LGS is higher than GGS”. Please simplify.
Response 18: The sentence has been modified in the revised manuscript.
Point 19: Line 440-442: Do the authors mean “the morphology and geometrical dimensions of LGS and AGS micro-scale structures are almost the same”?
Response 19: Yes. The sentence has been modified in the revised manuscript.
Reviewer 3 Report
The paper deal with the tribological properties of laser surface treated aluminum alloy with 3 different surfaces: linear groove, gridding groove and arc groove. The subject merits to be explored given the diffusion of applications in aluminum hindered by low surface properties.
The analysis mainly regards the friction coefficient and the wear rates while micro-hardness and chemical composition through EDS probe of SEM microscope are also reported.
In the analysis of experiments the experimental errors is not included, often data seem to be a single measurement (see for instance hardness in Figure 5 or all the points in wear rates and friction coefficients shown in several figures). Replications nor in the tests neither in the measurements are given, so it is difficult and source of mistake to compare single value and infer general conclusion. For instance in Figure 16 is difficult to infer statistically significant difference between 0.396 and 0.405 (friction coefficient) as the experimental error is not known. T
The authors have to improved the quality of the analysis giving evidence of the experimental errors and performing comparison between different groove surfaces based on the the analysis of variance rather than the analysis of avarage or single measurement points.
Moreover, the article is excessively long and de-focused despite the structure is good.
Lastly, English must be revised because a lot of errors are found (See for instance trobological instead of tribological).
Author Response
Point 1: In the analysis of experiments the experimental errors is not included, often data seem to be a single measurement (see for instance hardness in Figure 5 or all the points in wear rates and friction coefficients shown in several figures). Replications nor in the tests neither in the measurements are given, so it is difficult and source of mistake to compare single value and infer general conclusion. For instance in Figure 16 is difficult to infer statistically significant difference between 0.396 and 0.405 (friction coefficient) as the experimental error is not known.
Response 1: Thank you for your suggestion. Actually, every test such as hardness, friction coefficient and wear rate was carried out more than 4 times to ensure the reliability of experimental data. The standard deviation of the data was calculated and added in all graphs in the revised manuscript.
Point 2: The authors have to improve the quality of the analysis giving evidence of the experimental errors and performing comparison between different groove surfaces based on the analysis of variance rather than the analysis of average or single measurement points.
Response 2: Thank you for your suggestion. The performing comparison between different surfaces based on the analysis of standard deviation were discussed in the revised manuscript.
Point 3: the article is excessively long and de-focused despite the structure is good.
Response 3: Thank you for your comment. The manuscript has been re-edited.
Point 4: English must be revised because a lot of errors are found (See for instance trobological instead of tribological).
Response 4: Thank you for your comment. The English and grammar of the manuscript has been re-edited by MDPI language editing service. According to our investigation, numerous research articles use “tribological” but not “trobological”. Thus, “tribological” was not replaced by “trobological” in the revised manuscript.
Round 2
Reviewer 2 Report
Though the authors responded to the queries, the vocabulary and sentence constructions should be improved significantly.
Reviewer 3 Report
The authors have slightly improved the paper in agreement with the remarks. Now Iis worth to be published.